# Comparative Study of the Reinforcement Type Effect on the Thermomechanical Properties and Burning of Epoxy-Based Composites

Kamila Salasinska [1,*], Mateusz Barczewski [2], Joanna Aniśko [2], Aleksander Hejna [3] and Maciej Celiński [1]

1   Central Institute for Labour Protection–National Research Institute, Department of Chemical, Biological and Aerosol Hazards, Czerniakowska 16, 00-701 Warsaw, Poland; maciej.celinski@ciop.pl
2   Institute of Materials Technology, Poznan University of Technology, Piotrowo 3, 61-138 Poznan, Poland; mateusz.barczewski@put.poznan.pl (M.B.); joanna.anisko@put.poznan.pl (J.A.)
3   Department of Polymer Technology, Gdansk University of Technology, Narutowicza 11/12, 80-233 Gdansk, Poland; aleksander.hejna@pg.gda.pl
*   Correspondence: kamila.salasinska@ciop.pl

**Abstract:** Aramid (AF), glass (GF), carbon (CF), basalt (BF), and flax (FF) fibers in the form of fabrics were used to produce the composites by hand-lay up method. The use of fabrics of similar grammage for composites' manufacturing allowed for a comprehensive comparison of the properties of the final products. The most important task was to prepare a complex setup of mechanical and thermomechanical properties, supplemented by fire behavior analysis, and discuss both characteristics in their application range. The mechanical properties were investigated using tensile and flexural tests, as well as impact strength measurement. The investigation was improved by assessing thermomechanical properties under dynamic deformation conditions (dynamic mechanical–thermal analysis (DMTA)). All products were subjected to a fire test carried out using a cone calorimeter (CC).

**Keywords:** epoxy; composite; mechanical properties; fire behavior; inorganic fibers; basalt; glass fibers; carbon fibers; aramid; flax; fabrics

## 1. Introduction

The increasing requirements imposed on construction materials by the automotive, aviation, and civil engineering industries make it necessary to introduce new material solutions with high strength and, at the same time, significantly lower density than the metal materials used so far [1]. The most widely used as high-performance materials are fiber-reinforced polymers (FRP), particularly those based on thermoset polymers such as polyester and epoxy resins [2,3]. The possibility of using polymers with a high degree of cross-linking and reactive towards fillers allows obtaining structures with high mechanical strength and chemical resistance, with high adhesion at the polymer-filler interface [1,4,5]. The popularity of the thermoset composites in the form of laminates reinforced with long fibers is due to the possibility of producing high mechanical performance products without the necessity of expensive technological equipment use. The hand wet lay-up method, vacuum-assisted resin transfer molding (VARTM), or resin infusion allows obtaining thin-walled composite structures with good quality and relatively low amount of structural defects [6,7].

When selecting the appropriate reinforcement fibers, the fibers themselves' strength properties should be taken into account, and additional technological aspects, including wettability by the selected thermoset resins, thermal resistance, and density. All these features contribute to the product's final properties and the choice of the manufacturing technique. In recent years, even if the price criteria are still crucial at the stage of project and manufacturing on the large-scale, more attention has been paid to the sustainability of individual components that make up the final carbon and water footprint of a product [8,9].

Among the long fibers used in the production of laminates with a matrix of thermoset polymers, the most frequently used ones include glass fibers (GF), mostly S-glass and E-glass, depending on final purpose, i.e., tensile strength or chemical durability, respectively [6]. Manufacturing of the epoxy-based composites using GF fabrics allows achieving low-priced, and high-performance products [10–12]. To increase the mechanical strength and durability of epoxy laminates, it is common practice to replace them with carbon fibers (CF) [13–15]. They are not only resistant to stress corrosion and stress rupture failures like glass fibers but gain an advantage over other used reinforcements by additional functionality such as good electrical and heat conductivity [16]. This translates directly to the beneficial properties of composites developed with their participation [17–19]. Although basalt fibers (BF), similar to GF, often have more than 50% $SiO_2$, their properties are much superior. Moreover, their production is characterized by lower energy consumption and less complicated manufacturing processes [20]. BF's high strength, chemical, and thermal stability led to the wide application as a reinforcing material in polymer composites. The basalt fibers are well known for their excellent impact resistance; therefore, they are often used to hybridize laminates containing natural or synthetic fibers [21,22]. Synthetic fibers used as reinforcements are mainly made of heat-resistant and high-performance polymers such as polyamide or aramid. Aramid fibers (AF), showing properties comparable to inorganic fibers, are used in all applications where it is essential to reduce the final product's weight [20,21]. Even though aramid fibers show a low adhesion to polymers compared to inorganic or carbon fibers [23], it has been shown based on Mai and Castino studies [24] that surface modification can significantly improve the interfacial interactions at the polymer-filler interface. In recent years, researchers have been very interested in using natural fillers to produce polymeric composites [25,26]. The introduction of long or short fibers from such plants as flax [27], hemp [28], jute [29], kenaf [30], banana [31,32] allows increasing the sustainability of the final composite in comparison to those reinforced with inorganic fibers. This is mainly due to reducing the constraints on disposal in the combustion process and sometimes the lower energy needed to produce the fiber filler itself [33]. Considering the high availability in Europe as well as one of the highest tensile strength (700 MPa) among natural fibers, the use of flax fibers (FF) seems to be particularly justified [34]. A significant limitation for natural fibers is their high hydrophilicity and porous structure, which directly translates into porosity in the final composite products' structure [35,36].

In the case of fiber-reinforced composites intended for the production of high-strength structures, from the point of view of industrial application, the thermomechanical resistance and flame resistance of the material are the criterion of no less importance than the strength itself [37,38]. Each introduction of fibers into the polymer matrix, even if the matrix itself's cross-link density does not change, causes a reinforcing effect that allows achieving improved stiffness at elevated temperature in comparison to pure polymer. In the case of natural fibers use, while the reinforcing efficiency is still high [39], the use range is lower, mainly due to the lignocellulosic filler's limited thermal stability compared to inorganic or carbon fibers [40]. CF reinforced epoxy composites are characterized by very high stiffness at elevated temperature values, higher than BF or GF [41]. Despite BF and GF's similar chemical structures, epoxy composites produced with a comparable fiber fraction of BF are usually characterized by higher storage modulus values at elevated temperatures [42]. Chinnasamy et al. [43] presented the thermomechanical behavior of epoxy glass-aramid composites and showed that synthetic fibers' additions caused an increase in stiffness at elevated temperatures.

The introduction of thermally stable fillers to the polymer partially reduces the composite material's caloric value due to the partial replacement of highly flammable polymer. However, the use of long inorganic (BF, and GF) or carbon (CF) fibers does not result in obtaining fire retardant materials [44]. Additionally, in the case of splitting of composites modified with synthetic (AF) or natural (FF) fibers, a much more significant amount of smoke is emitted [37,45]. However, it should be mentioned that despite the organic

origin of aramid fibers, their introduction into the epoxy matrix allows achieving lower flammability than for composites reinforced with glass or graphite fibers [45].

The composite's mechanical properties depend on reinforcing fibers' mechanical performance and the structure of the used fabric, its amount, the propensity of its saturation by polymeric matrix, interfacial adhesion, and structural defects resulting from applied manufacturing technology [46]. Interestingly enough, while the literature offers much information on epoxy composites' properties, there is no comprehensive data set for simple composites that differ in terms of the type of fiber used. Therefore, the data presented in this paper can be a source of engineering reference data in the case of the production of more complex composite systems. The study aimed to make a critical analysis of the performance properties of five types of epoxy composites manufactured by the hand lay-up method and reinforced with fibers of various origins, i.e., glass, basalt, carbon, aramid, and flax. The mechanical and thermomechanical properties' evaluation was related to changes in the structure and completed with the carried out burning behavior carried out using a cone calorimeter.

## 2. Experimental

### 2.1. Materials and Sample Preparation

The polymeric matrix used in the studies consists of epoxy resin based on bisphenol A diglycidyl ether, RenLam LY 113 (viscosity 580 mPa·s at 25 °C and density 1.16 g/cm$^3$), and Ren HY 97-1 used as a hardener (viscosity 20 mPa·s at 25 °C and density 0.95 g/cm$^3$) from Huntsman.

Five types of fabrics made of different fibers, such as aramid (AF), glass (GF), carbon (CF), basalt (BF), and flax (FF), have been used for manufacturing of epoxy-based laminates. The mechanical properties of used in this study fibers are presented in Table 1.

**Table 1.** Mechanical properties of the single fibers used in the study.

| Fiber Type | Elasticity Modulus (GPa) | Tensile Strength (GPa) | Reference |
|:---:|:---:|:---:|:---:|
| AF | 102 | 2.8 | [47,48] |
| GF | 76 | 1.4–2.5 | [33] |
| CF | 354–375 | 5.5–5.8 | [49] |
| BF | 89 | 2.8 | [20] |
| FF | 30–38 | 0.45–1.5 | [33,50] |

Aramid fabric (AF) delivered by P.P.H.U. SURFPOL (Poland) with a twill weave (twill 2/2) and a weight of 300 g/m$^2$ was made using Tex 240 fibers. Sewn glass fabric made of E-glass (GF) BIAX 400 g/m2 manufactured by Saertex GmbH & Co. KG (Germany) is a two-way fabric (+45/−45°) with a grammage of 411 g/m$^2$. Carbon fibers (CF) as sewn carbon fabric trade name BIAX 400 g/m2 were delivered by Saertex GmbH & Co. KG (Germany). It is a two-way fabric (+45/−45°) with a grammage of 410 g/m$^2$. Basalt Multiaxial Fabric (BF) BAS BI 450 by Basaltex NV (Belgium) is a two-way sewn fabric (+45/−45°) with a grammage of 464 g/m$^2$. For the manufacturing of the composites, a flax fabric (FF) manufactured by Safilin (France) with a twill weave (2/2 twill) and a grammage of 500 g/m$^2$, made of Tex 400 fibers was used. Figure 1 presents the photographs of all used fabrics, while Figure 2 shows the images of single fibers under magnification of ×500 and the average fiber thickness.

Manufacturing of the laminates was realized using the constant resin-to-curing agent ratio of 100:30 by weight. The compositions were mechanically mixed using proLAB 075 stirrer with a rotational speed of 2000 rpm for 3 min under subatmospheric pressure. The six-layer laminates with dimensions of 330 × 330 mm$^2$ were manufactured by hand lay-up method. Additionally, the reference samples from unmodified epoxy resin (EP) were formed by casting into Teflon molds with dimensions of 140 × 10 × 4 mm$^3$ as well as silicon one with dimensions of 100 × 100 × 4 mm$^3$. After the forming process, samples were cured

at room temperature for 72 h and post-cured for the next 3 h at 70 °C using Goldbrunn 1450 vacuum dryer. For each kind of composite, the same amount of epoxy composition (approximately 390 g) has been used. In the other part of the study, the names of selected composites names will be marked with appropriate abbreviations corresponding to the fibers used. Based on the known amount of ingredients used and their characteristics, it can be determined that the composites contain the following amount of filler AF 33.4 wt% (30.1 vol%), GF 40.8 wt% (26.4 vol%), CF 40.7 wt% (32.2 vol%), BF 43.7 wt% (28.9 vol%), and FF 45.6 wt% (40.8 vol%).

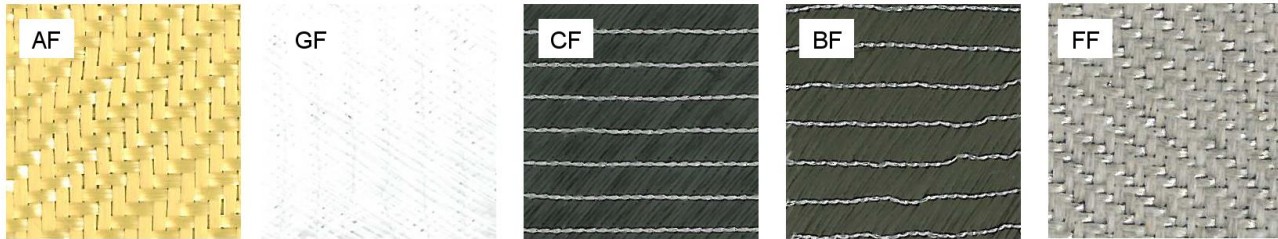

**Figure 1.** Photographs of fabrics used for the manufacturing of the epoxy-based laminates.

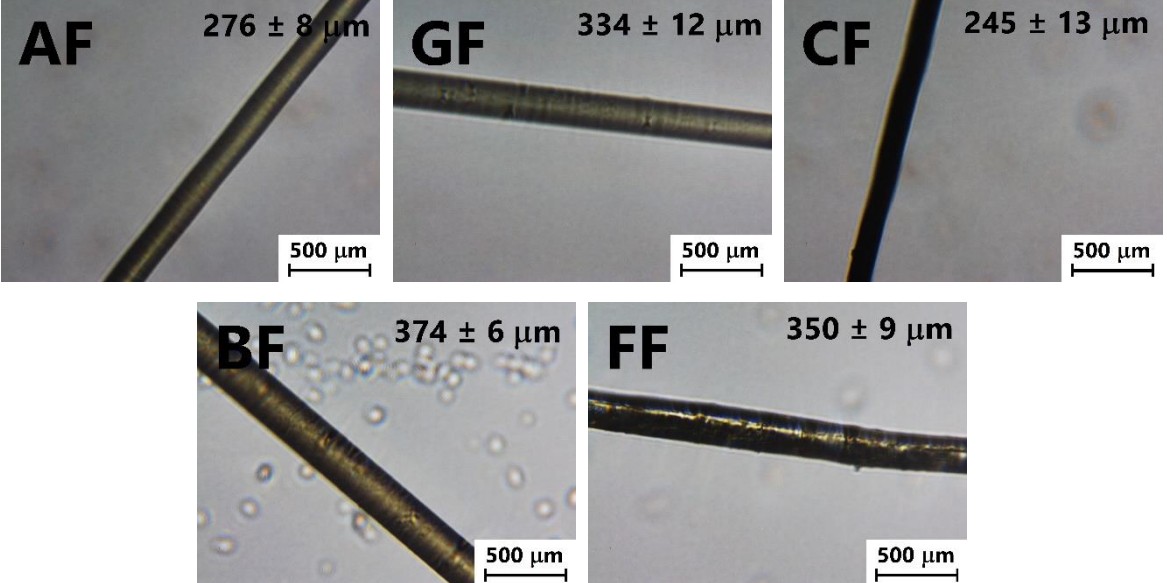

**Figure 2.** Microscopic images of single fibers used as reinforcement with a diameter measurement.

### 2.2. Methods

The Fourier transform infrared spectroscopy (FTIR) measurements were realized using a spectrometer Jasco FT/IR-4600 (USA), at room temperature (23 °C) in a mode of Attenuated Total Reflectance (FTIR-ATR). A total of 32 scans at a resolution of 4 cm$^{-1}$ was used in all cases to record the spectra.

The specific weight of fillers and composites was determined using Ultrapyc 5000 Foam gas pycnometer from Anton Paar (Austria). The following measurement settings were applied: gas—nitrogen; target pressure—0.124 MPa; flow direction—reference first; temperature control—on; target temperature—20.0 °C; flow mode—monolith; cell size—10 cm$^3$; preparation mode—flow, 0.5 min; the number of runs—10.

The mechanical behavior of the composites was realized using static tensile and flexural tests. The measurements were performed by ISO 527 and ISO 178 standards with a Zwick/Roell Z020 (Germany) universal testing machine at room temperature. The measurements were conducted at a crosshead speed of 5 mm/min. For each series, five specimens were tested.

The impact strength of the unnotched samples was examined by the Charpy method according to ISO 179 standard at 25 °C. The Zwick/Roell HIT 25P (Germany) impact tester with a 5 J hammer was applied for the measurement, and the peak load was determined as the maximum force ($F_{max}$). For each series, five specimens were tested.

Microscopic observations of the composites structures were performed using an optical microscope Opta-Tech SK Series microscope (Poland). The obtained pictures were digitally edited and registered using dedicated software.

The dynamic mechanical–thermal analysis (DMTA) was conducted in torsion mode using Anton Paar MCR 301 rheometer with the SRF measuring system. Investigations were carried out with a constant frequency of 1 Hz and a strain of 0.01%. The measurements were realized in a 25 °C to 200 °C temperature range with a heating rate 2 °C/min.

The fire behavior was determined by a cone calorimeter from Fire Testing Technology Ltd., following ISO 5660. The samples with dimensions of $100 \times 100$ mm$^2$ were wrapped in aluminum foil and placed on the holder in a steel frame, providing a surface area of 88.4 cm$^2$. All specimens were irradiated horizontally at a heat flux of 35 kW/m$^2$. For each series, six specimens were tested.

## 3. Results and Discussion

The chemical structure of used fibers and epoxy composites reinforced with various fillers were assessed by the FTIR method. The measurements were realized in FTIR-ATR mode, and the analysis is concerning the chemical structure of the fibers and composite surface. Figure 3 presents the FTIR spectra of used reinforcements. The characteristic absorption bands for each fiber may be observed. The complex FTIR spectra of AF consist of as listed prominent absorption bands 3300 cm$^{-1}$ (N-H stretching), 1638 cm$^{-1}$ (C=O stretching, Amid I), 1536 cm$^{-1}$ (coupled C-N stretching and in-plane N-H bending, Amid II), 1400 cm$^{-1}$ (primary amine salt), 1306 cm$^{-1}$ (C-C, C-N group motion and N-H vibrations, Amid III), 1103 cm$^{-1}$ (C-H in-plane bending), 892 (C-H out of plane), 862 cm$^{-1}$ (C-H out of plane bending), and 725–844 cm$^{-1}$ (N-H and aromatic C-H out of plane bending) [51–54]. For both inorganic fillers, i.e., BF and GF, the most distinct wide absorption band with a maximum at 897 cm$^{-1}$ was originated from SiO stretching mode [55]. For glass fiber, additional absorption bands were observed resulting from the silane coupling agent, including at 1715 cm$^{-1}$ (C=O stretching) and 720 cm$^{-1}$ (the presence of residual water) [55,56]. The lack of absorption bands in considered wavelength is characteristic of CF [57]. For FF fiber, several absorption bands were denoted, including 3337 and 3296 cm$^{-1}$ (O-H stretching band), overlapping 2900 and 2896 cm$^{-1}$ (C-H stretching and C-H$_2$ symmetrical stretching), 1713 cm$^{-1}$ (C=O carbonyl stretching of hemicellulose), 1620 cm$^{-1}$ (water absorbed in cellulose), 1432 and 1372 cm$^{-1}$ (-CH$_3$ asymmetric and C-H symmetric deformations from lignin), 1335 cm$^{-1}$ (aromatic ring of the cellulose), 1101 cm$^{-1}$ (C-O-C symmetric glycosidic stretch), 1052 cm$^{-1}$ (C-OH stretching vibration of cellulose backbone), and 896 cm$^{-1}$ (phase ring stretching from the cellulose backbone) [58–60].

Figure 4 shows the characteristic spectra for epoxy-based composition after curing. The absorption band in the 3700–3200 cm$^{-1}$ range corresponds to hydrogen bonds (-OH groups) [61]. Moreover, the following peaks from the epoxy matrix can be distinguished on the spectra: at 2800–3000 cm$^{-1}$ from the -CH$_2$ and -CH$_3$ symmetric stretching band; at 1500–1600 cm$^{-1}$ from the Ar-C = C-H stretching band; at 1150–1300 cm$^{-1}$ from -C-C-O-C- stretching band, at 1000–1100 cm$^{-1}$ from -C-O-C- stretching band and at 800–850 cm$^{-1}$ from the aromatic ring. Lack of additional peaks resulting from the reinforcing fillers suggests good saturation of the fabrics by polymeric matrix and creating a coherent layer at the composite's surface, which will have a beneficial impact on the performance, including water absorption crucial for hydrophilic flax fibers. The degree of cross-linking of the epoxy composition may be estimated based on the absorption bands decay of epoxy groups at a wavelength of 915 cm$^{-1}$, which are characteristic for group valence vibrations in uncured epoxy resin [62,63]. The disappearance of the peaks in this band of the FTIR spectrum, in the case of all tested materials, proves that epoxy matrix has been fully cured, which

confirms proper realization of the manufacturing process and lack of negative effect of any of the used fillers on the thermoset polymeric matrix.

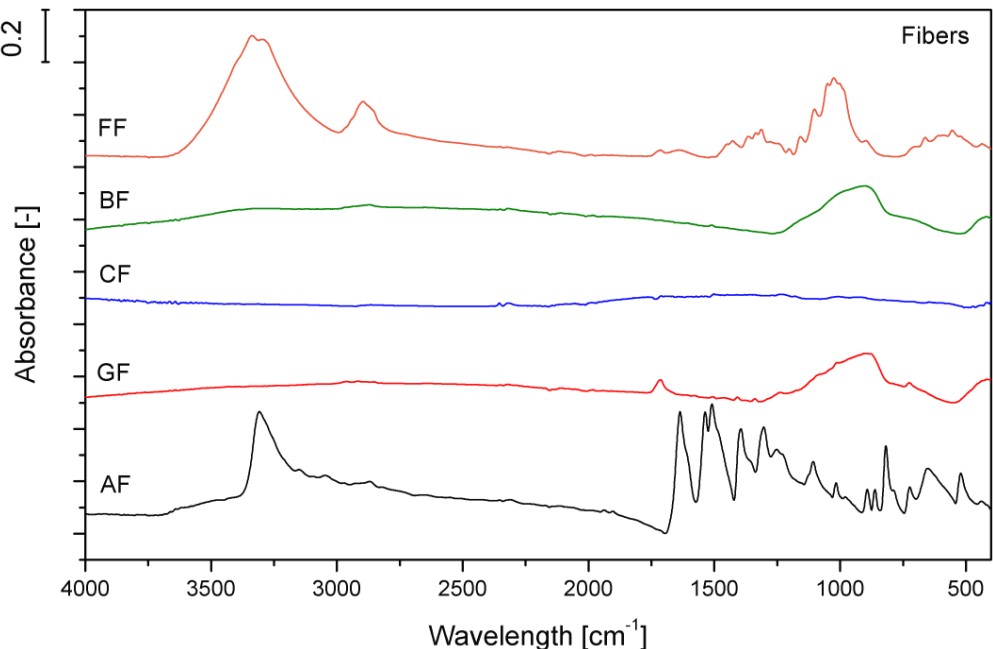

**Figure 3.** FTIR-ATR spectra of fibers used in the study.

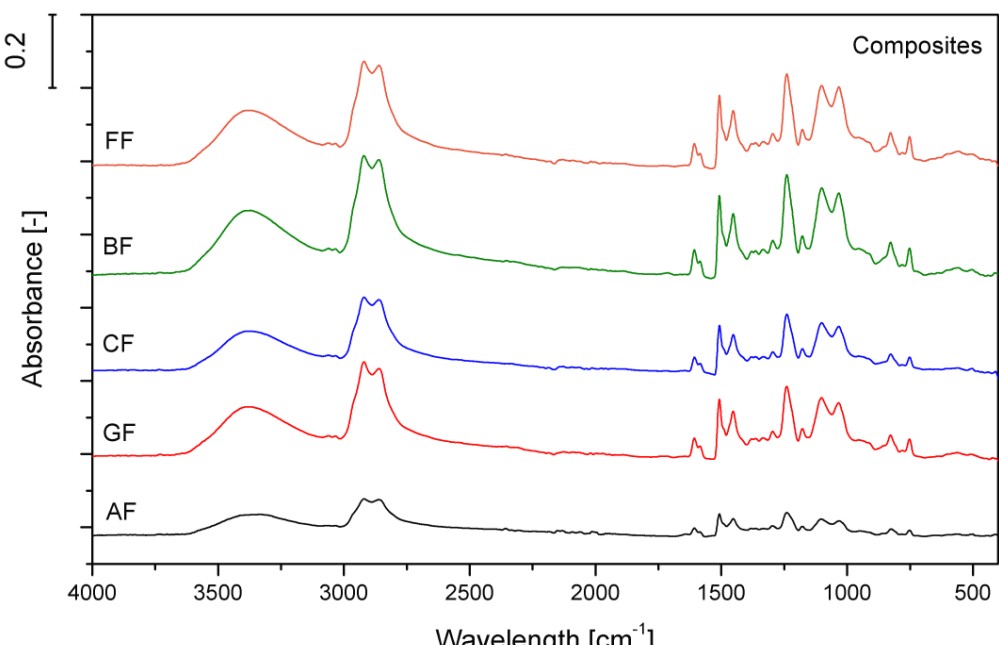

**Figure 4.** FTIR-ATR spectra of EP laminates.

Figure 5 shows the relationship between the density of applied fabrics and manufactured composites. Presented values are related and proportional, and any deviations result from the differences in the chemical structure of fabrics and possible porosity. Such an effect may be associated with the inaccuracies during hand lay-up lamination, but mostly with the differences in particular fabrics' chemical structure, presented in Figure 6.

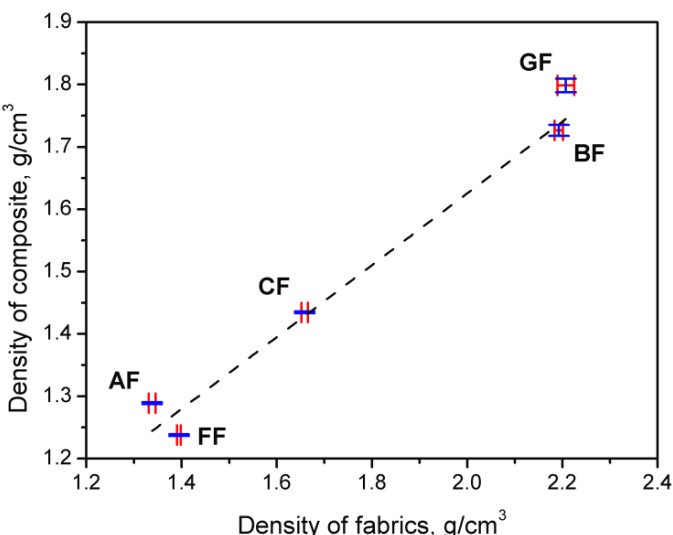

**Figure 5.** The density of composites and density of fabrics.

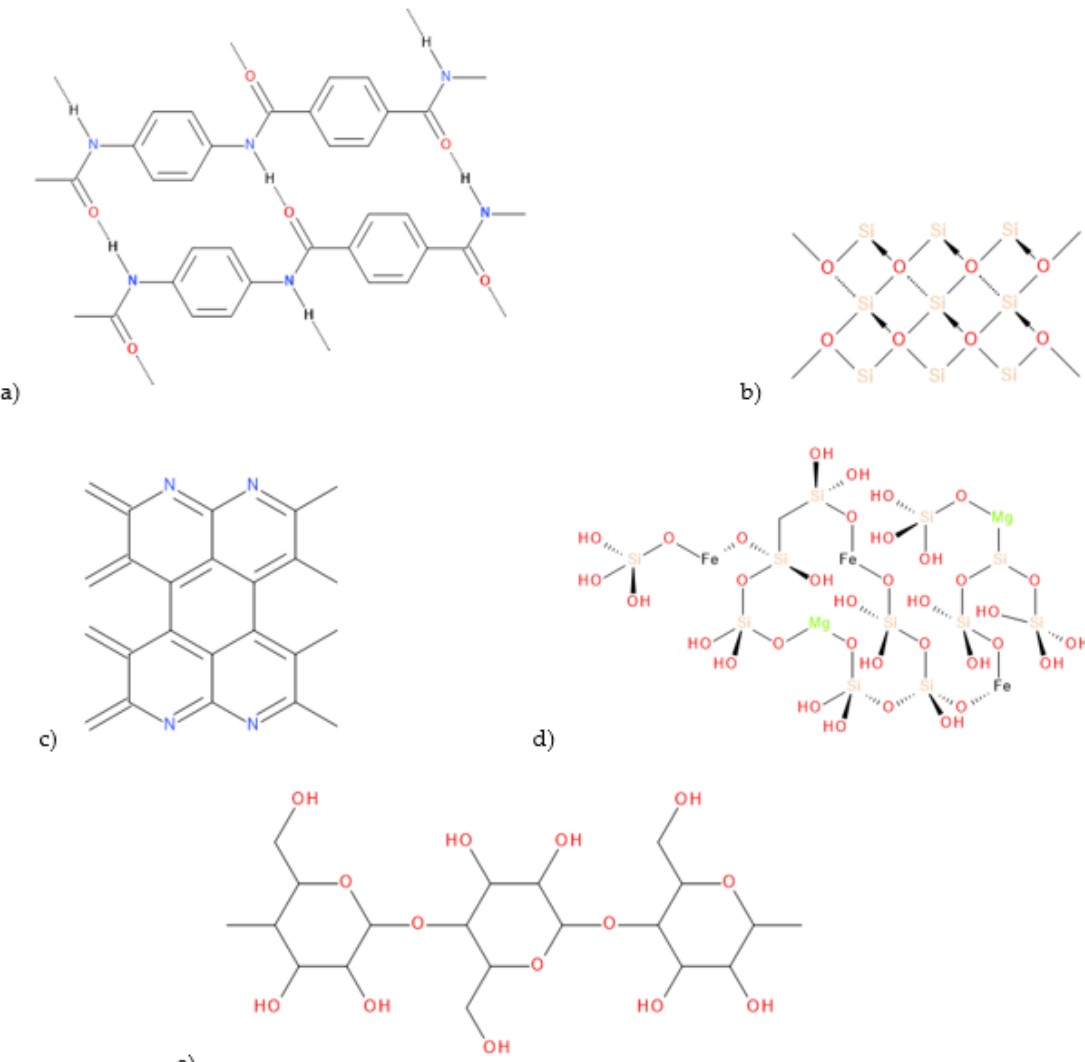

**Figure 6.** Chemical structure of the fibers: aramid fiber (AF) (**a**), glass fiber (GF) (**b**), carbon fiber (CF) (**c**), basalt fiber (BF) (**d**), flax fiber (FF) (**e**).



According to the literature data, basalt fibers are mostly composed of silicon dioxide (49–51%), aluminum oxide (14–16%), iron oxides (7–13%), calcium oxide (10%), and magnesium oxide (6–16%), with minor shares of sodium, potassium, titanium, and manganese oxides [64]. As shown in Figure 6, the flax fibers may be described as the most hydrophilic due to multiple hydroxyl groups' presence in their structure. Aramid fibers also contain a significant amount of relatively polar amine groups; however, they are all connected with carbonyl oxygens with hydrogen bonds. Such an effect is one of the reasons for the exceptional properties of aramid fibers [65]. As a result of these differences, the laminate containing flax fabric was characterized by the void content significantly higher than for other samples (data point in Figure 5 lies below the line showing proportionality of fabrics' and composites' densities).

Figure 7 summarizes the mechanical tests of layered composites and epoxy resin as a reference carried out in the tensile test. Separate graphs show the averaged values of modulus of elasticity, tensile strength measured at break of the sample, elongation at break, as well as selected examples of tensile stress as a function of relative strain. The highest tensile strength and stiffness characterized composites reinforced with aramid fabric. The Young's modulus and tensile strength values were above 12 GPa and 350 MPa, respectively. The most unsatisfactory mechanical performance from all composites was observed for glass fiber reinforced composites. In turn, composites modified with basalt fiber showed stiffness and tensile strength higher by 1 GPa and 45 MPa than GF composites. Despite the fact that the composites reinforced with carbon fibers were characterized by increased stiffness compared to composites with inorganic fibers, this value was still 30% lower than the average values recorded for AF. CF composites' immediate strength was between BF and GF's strength and was similar to the only series of the composite reinforced with natural fibers (FF). Although composites reinforced with FF showing a tensile strength higher than those reinforced with GF, Young's modulus of the natural fiber composite was the lowest among all considered materials. An interesting relationship was observed in the analysis of mean values of elongation at break. The composites containing inorganic fibers, i.e., GF and BF, were characterized by the highest values. Increased elongation at break characterized composites containing glass and basalt fibers, that results from both a lower share of fibers in the load transfer and lower volume content of filler, as well as interlaminar propagation of the breakage leading to the extension of the damaged area in the sample (Figure 8). Observed in the stress vs. strain curves change in the materials' fracture characteristics was also noted for CF composite; however, the elongation at break was much lower than for GF and BF, mostly due to a higher amount of the fibers in the composite structure. A typical brittle fracture pattern is observed in the case of low-density fiber-reinforced materials (AF and FF).

Figure 8 shows the composite samples' profile view after the tensile test, taken using an optical microscope. Based on the images of composites except for FF, referred to the example of the stress–strain curve, it can be stated that cracking at the interface of matrices occurs at the point of maximum stress and then propagates along the fibers and, as a result, appears at the external surface of the sample [66]. In summary, the cracking mechanism of the samples subjected to stretching in the samples AF, BF, CF, and GF is characterized by a dominant debonding effect, leading to a pull-out phenomenon of the fibers [66–68]. A different cracking mechanism was observed only in FF composites, which ultimately broke during the tensile test and showed no distinct interlaminar breakages. This may be related to the lowest mechanical strength of the natural fibers themselves.

**Figure 7.** Mechanical properties of EP and EP-composites obtained from tensile test.

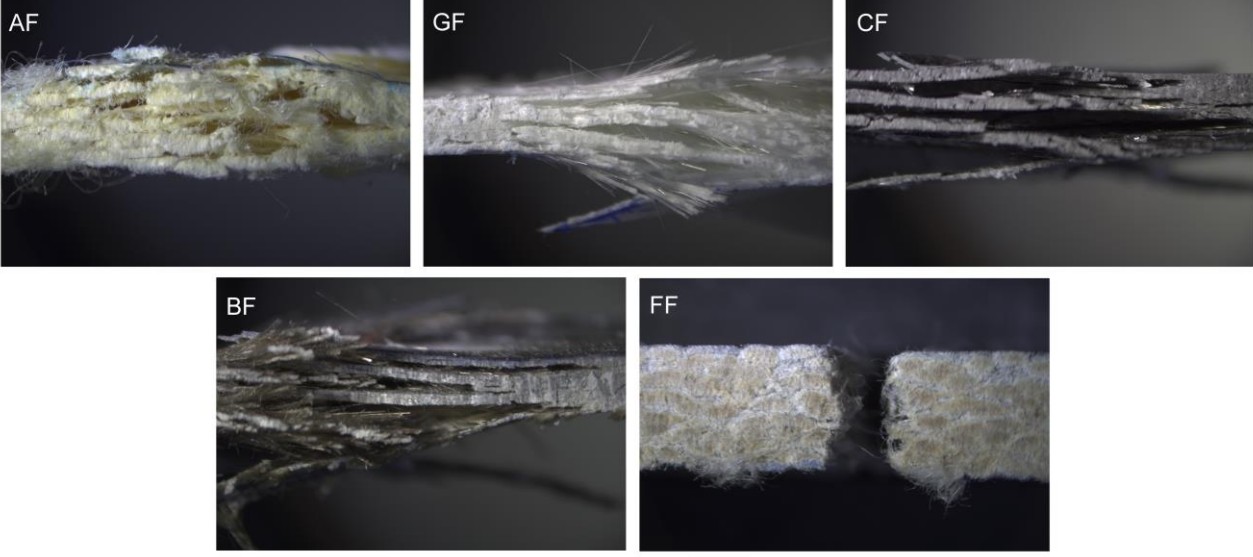

**Figure 8.** Profile view of the composite samples after tensile test by an optical microscope.

The mechanical properties of unmodified epoxy and its composites with different types of fibers assessed in three-point bending test, realized up to strain calculated according to ISO 178 standard, are shown in Figure 9. Based on the averaged flexural modulus and flexural strengths, it can be concluded that in the case of loads performed in the bending system, CF reinforced composites showed the most favorable properties. For these materials, both the highest flexural modulus value (12.5 GPa) and the flexural strength determined at the maximum force value (185 MPa) were recorded. The flexural modulus, in the case of AF and CF samples, was comparable. Simultaneously, the significantly more notable standard deviation of the composites reinforced with synthetic fibers indicates a less homogeneous structure of the composite, worse plain mechanical adhesion, or the presence of additional structural defects in the form of voids and pores. The lowest value of stiffness was shown by samples filled with flax fibers, while of bending strength by samples with GF. It can be stated that the strength of the composites, in comparison to the results obtained in the uniaxial tensile test, except for the CF sample, did not significantly change. The difference between GF and BF was only 24 MPa.

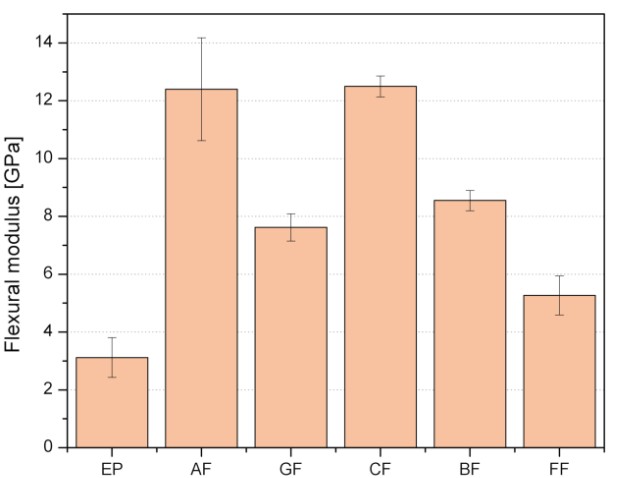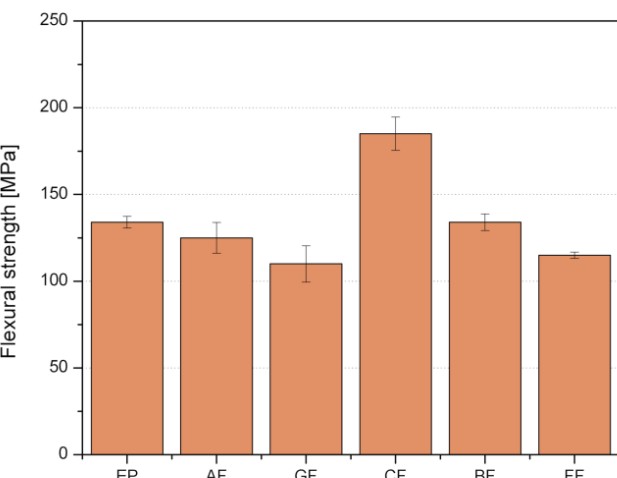

**Figure 9.** Mechanical properties of EP and EP-composites obtained from the flexural test.

The course of the processes of damaging the structure of fiber-reinforced composite materials is a complex mechanism resulting from interlaminar damage modes, which include the phenomenon of matrix and fiber cracking and the occurrence of interlayer damage, mainly delamination. Delamination can result from impact, bearing loading of the bonded joint, or any other source of significant interlayer stress [69]. The impact strength of fabric-reinforced epoxy composites was assessed by the Charpy method without the notch. The averaged results of the impact toughness and the associated maximum force (Fmax) recorded during the tests are summarized in Figure 10. Additionally, to complete the sample crack analysis, a profile view of selected samples after measurements, made using optical microscopy, is presented in Figure 11. The BF and CF reinforced composites showed the highest impact strength, while the lowest-natural ones, excluding unmodified EP. The results are consistent with the literature data [70]. According to Papa et al., natural basalt fibers' introduction into the composites allowed to increase their impact resistance [71]. In turn, the most significant force accompanying the impact was recorded for the FF. It should be emphasized that Fmax was related to the thickness of the composite. Brittle fracture is characterized by a sharp excess of the yield point and a subsequent rapid rupture of material structure [6]. The highest Fmax recorded for the unfilled epoxy resin result from typical for highly-cured polymers abrupt brittle fracture. Considering the photographs of fractures in composite samples presented in Figure 11, it can be stated that it was brittle in the event of a crack. Only FF composites showed complete loss of structure after impact load, while, in the case of GF and CF, delamination was observed, more visible in CF's case. For CF specimen shows the different fractographic feature in

comparison to rest of the laminates, it showed extensive longitudinal splits due to the local fracture in this region prior to the arrival of the main crack front, which resulted in higher amount of the secondary cracks in final fracture zone [72]. GF, CF, and FF samples showed multiple fiber breakage and visible fiber pull-out effect, correlated with lower interfacial strength [70]. After impact, the AF samples showed permanent deformation, although the reinforcing fabric structure was not damaged. In composites made with basalt fibers, only slight damage to the composite's outer surface was noted. The BF composites were not delaminated after impact load.

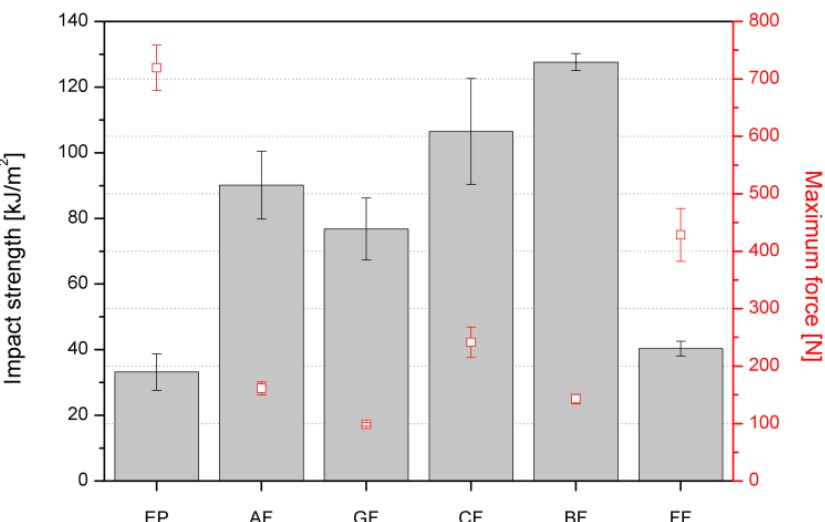

**Figure 10.** Charpy impact strength and corresponding measured Fmax during the test of the EP and EP-based composites.

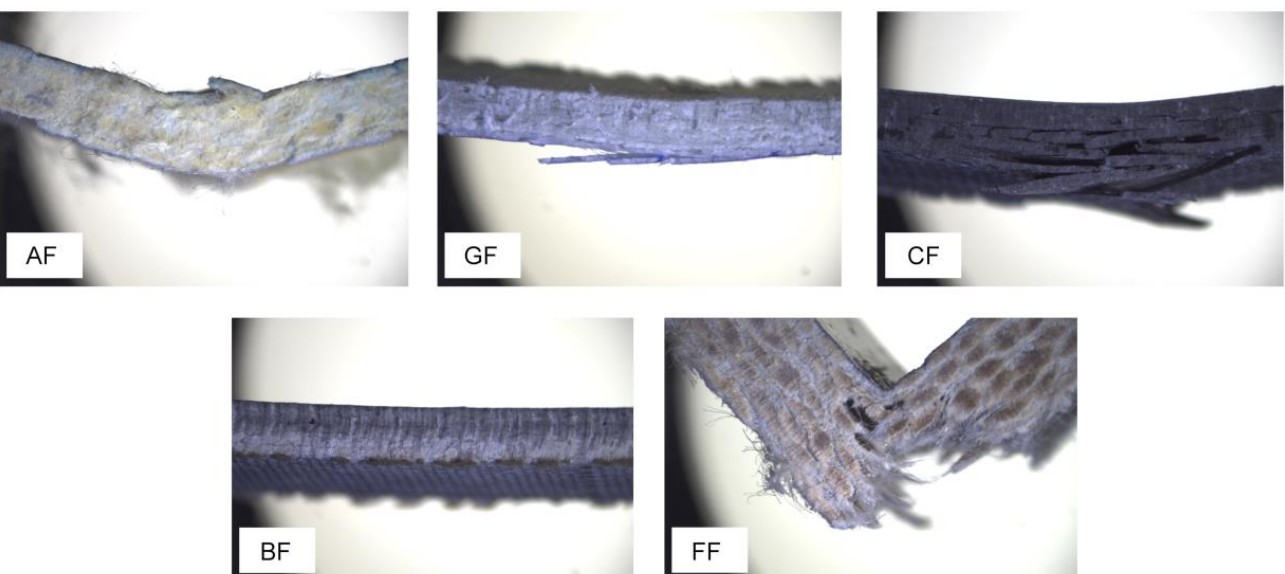

**Figure 11.** Profile view of the samples after Charpy impact test by an optical microscope.

Brittleness is a multicriterial parameter proposed by Brostow [73], which allows qualitative assessment of the behavior of polymeric materials and their composites [73–75]. It can be calculated indirectly by referring to the added measurements from two different techniques, i.e., the results of static stretching and DMTA. The following formula can describe brittleness (B):

$$B = 1/(\varepsilon_b \cdot E') \tag{1}$$

where $\varepsilon_b$ is the elongation at break determined in tensile measurements, and E' is the storage modulus determined at the same temperature conditions and 1 Hz frequency by (DMTA). In this study, the values of storage modulus (G') determined in a torsion mode of dynamic mechanical analysis were applied to the Equation (1). In Table 2, the brittleness parameters for tested composites are presented. Although the calculated brittleness results are obtained from tensile measurements and dynamic analysis for non-destructive torsional deformations, results correlate with Charpy strength experiments. Samples with the lowest B value showed the highest impact resistance, and the FF composites, with the lowest impact load resistance, were characterized by a high B value, comparable to unmodified epoxy. The difference between the results of impact strength and brittleness of AF results from synthetic fibers' different behavior characteristics compared to natural and inorganic fibers.

**Table 2.** Thermomechanical parameters and brittleness of EP-based composites.

| Sample | $G'_{30°C}$ | $G'_{80°C}$ (Pa) | $G'_{140°C}$ | Tg (°C) | tanδ at Tg (-) | C (-) | B ($10^{10}$/Pa·%) |
|---|---|---|---|---|---|---|---|
| EP | $1.46 \times 10^9$ | $9.01 \times 10^8$ | $1.05 \times 10^7$ | 101 | 0.999 | - | 1.052 |
| AF | $1.05 \times 10^9$ | $1.23 \times 10^9$ | $1.27 \times 10^8$ | 101 | 0.385 | 16.81 | 1.180 |
| GF | $6.26 \times 10^9$ | $5.13 \times 10^9$ | $3.16 \times 10^8$ | 104 | 0.62 | 7.02 | 0.227 |
| CF | $13.25 \times 10^9$ | $11.13 \times 10^9$ | $5.72 \times 10^8$ | 103 | 0.597 | 6.00 | 0.138 |
| BF | $8.42 \times 10^9$ | $7.11 \times 10^9$ | $4.94 \times 10^8$ | 101 | 0.634 | 8.16 | 0.089 |
| FF | $1.73 \times 10^9$ | $1.29 \times 10^9$ | $1.07 \times 10^8$ | 106 | 0.387 | 8.60 | 1.051 |

The thermomechanical analysis results are presented in Figure 12 in the form of changes in storage modulus (G') and damping factor (tanδ) as a temperature function. Additional information about thermomechanical parameters, including the glass transition temperature (Tg) value, determined as the maximum on the tanδ (T) curve, is summarized in Table 2. Usually, the values of the elastic modulus correlate with the observed changes in the storage modulus. In the case under consideration, significant discrepancies between these methods were shown. The differences between the results occur mainly from the different volumetric content of the filler in composites and fabric structure (sewn/non-sewn), which resulted in different behavior of the samples during deformation with various geometries load.

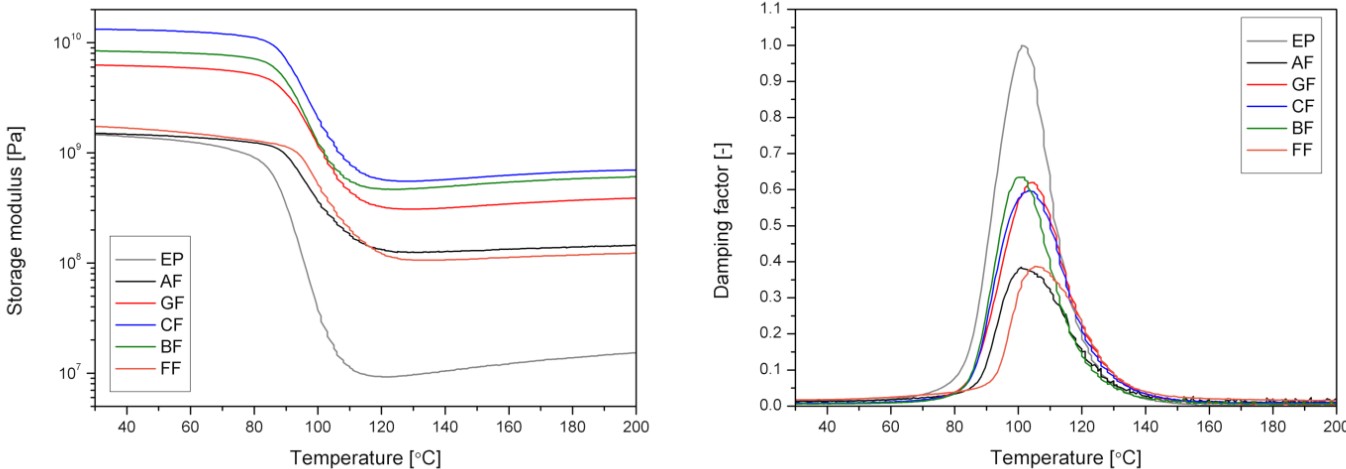

**Figure 12.** Storage modulus and damping factor vs. temperature curves of EP and EP laminates made using various fibers obtained by dynamic mechanical–thermal analysis (DMTA).

The course of the storage modulus and damping factor vs. temperature curves showed epoxy resin-based materials' typical behavior [76]. The highest stiffness understands as G' in the whole temperature range was observed for CF composite. Other inorganic

fiber-reinforced composites, i.e., BF and GF, showed, respectively, 63 and 45% of the CF $G'_{30°C}$. The difference between CF and BF or GF reinforced samples was lowered after reaching the polymeric matrix's glass transition. Materials containing aramid and flax fibers reveal comparable and significantly lower storage modulus valued in the whole temperature range than CF, BF, and GF. Moreover, the damping factor's maximum value may be translated into the ability to dissipate the mechanical energy and vibrations by material [77]. The composites containing synthetic and natural fibers showed significantly lower tanδ. The damping properties were much higher for materials containing inorganic and carbon fibers, and the recorded values differed insignificantly. This phenomenon may affect the higher volumetric content of the AF and FF in the composite structure and lower the natural fibers' stiffness.

Furthermore, all specimens showed a comparable value of Tg to unmodified EP. Lack of differences between materials may be an effect of a correctly realized post-curing procedure. Because the peak of tanδ curve represents the internal molecular motion of the polymer, but also the dissipation of energy at interfacial region [78], it can be supposed that the increased Tg in the case of FF may result from the more complicated mechanical interactions in the interface between the thermoset polymer matrix and organic fibers.

The effect of the used reinforcing fiber type on the composites' thermomechanical stability was also confirmed by the analysis "C" factor, which allows assessing the effectiveness of the filler on the polymeric matrix in a quantitative way. The values of C factor can be calculated according to the following Formula (2) [79]:

$$C = \frac{\left( G'_g/G'_r \right)_{composite}}{\left( G'_g/G'_r \right)_{matrix}} \tag{2}$$

where $G'_g$ and $G'_r$ are the values of storage modulus determined in the glassy state and in the rubbery state, after passing the glass transition of the material. It is assumed that the smaller the value of C factor, the greater the effectiveness of the filler on the polymeric matrix. The highest values were obtained for composites reinforced with CF and inorganic fibers, while the lowest for composites reinforced with AF.

In Table 3, results from cone calorimetry tests simulating a developing fire scenario with the use of a small sample [80] are presented. An increase in time to ignition (TTI) was observed for all composites, excluding materials with aramid fabric, and the highest value was recorded for GF. TTI values strongly depend on density, thickness, and conductivity of samples; therefore, the more reliable parameter is heat release rate (HRR) versus time [81], shown in Figure 13. The representative curve of the heat release rate of unmodified EP presents a peak heat release rate at the beginning of the burning at ca. 200 s. The curve consists of two main stages, the first of which gives the pHRR, and the second was a minor maximum. In the case of CF and FF, the opposite trend was observed, while, for AF, GF and BF, the first peak yielded pHRR, and then the significant flattening of the curves can be noticed. The use of fabrics resulted in a significant reduction in the pHRR compared to the unmodified resin, and the lowest results were determined for GF, AF, and BF. In the case of these materials, approximately the threefold decrease in the pHRR value was recorded. Another critical parameter used to predict a fire's development is the maximum average rate of heat emission (MARHE) [82]. The lowest values were obtained for composites with glass and aramid fabrics. Furthermore, in almost all composites, a reduction in total heat release (THR) was noted; the only exception was composite with natural fiber, for which THR were higher by 27% than EP.

**Table 3.** Cone calorimeter results of EP and EP-based composites.

| Sample | TTI, s | pHRR, kW/m² | MARHE, kW/m² | THR, MJ/m² | SEA, m²/kg | TSR, m²/m² |
|---|---|---|---|---|---|---|
| EP | 102 (8 [a]) | 1313 (109) | 385 (7) | 119 (3) | 606 (15) | 2969 (80) |
| AF | 92 (0) | 429 (229) | 177 (34) | 77 (10) | 557 (103) | 1855 (74) |
| GF | 133 (3) | 408 (15) | 187 (11) | 75 (1) | 662 (17) | 2099 (38) |
| CF | 119 (2) | 740 (9) | 263 (1) | 79 (1) | 623 (18) | 2078 (70) |
| BF | 105 (12) | 460 (28) | 201 (9) | 72 (1) | 617 (2) | 1819 (30) |
| FF | 129 (7) | 669 (143) | 274 (18) | 151 (5) | 423 (14) | 3220 (73) |

[a] The values in parentheses are the standard deviations.

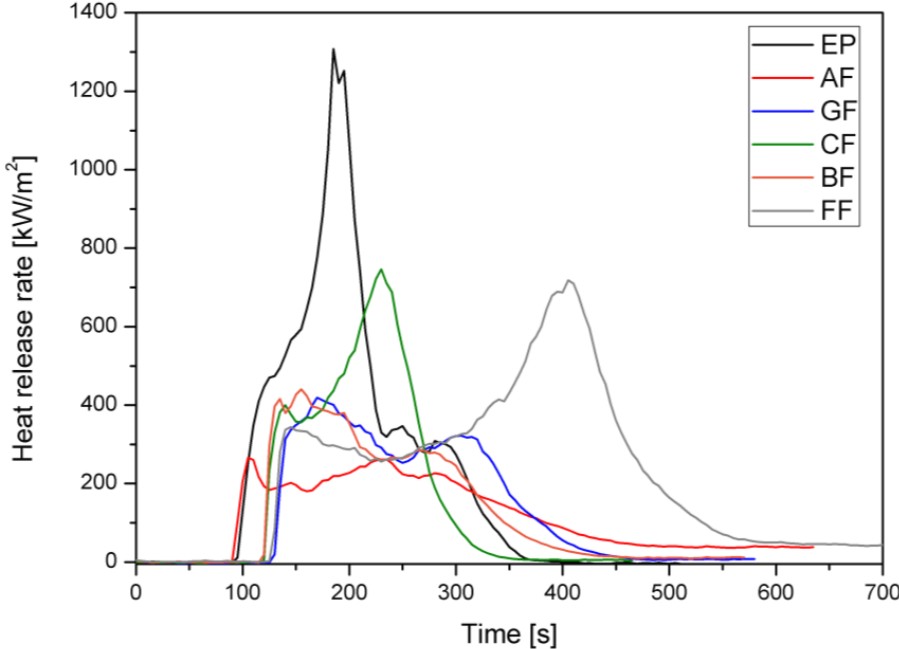

**Figure 13.** Heat rate release for EP and EP-based composites reinforced with various fibers.

The smoke that appears during a fire significantly reduces the ability to see and makes it difficult to carry out an effective evacuation action. The smoke emission based on tests performed using a cone calorimeter is determined by the specific extinction area (SEA), and the total smoke released (TSR). The SEA for the composites with glass, basalt, and carbon fabrics was slightly higher than the result obtained for the epoxy resin, and the highest value determined for BF was 12% higher than the unmodified polymer. On the other hand, in the case of composites with aramid and linen fabrics, there was a decrease in the analyzed parameter, and the lowest value was noted for the FF (reduction by 30%). Similarly, a decrease in the total amount of generated smoke was observed; however, contrary to the SEA, the highest values were found for composites with flax. The reduction in the intensity of burning and smoke emission resulted from the lower amount of resin, which is the most flammable component.

Figure 14 shows the photographs of composites and unmodified resin after cone calorimeter tests. The EP sample was almost completely burnt, while in the case of the composites, the residue consisted mostly of reinforcement.

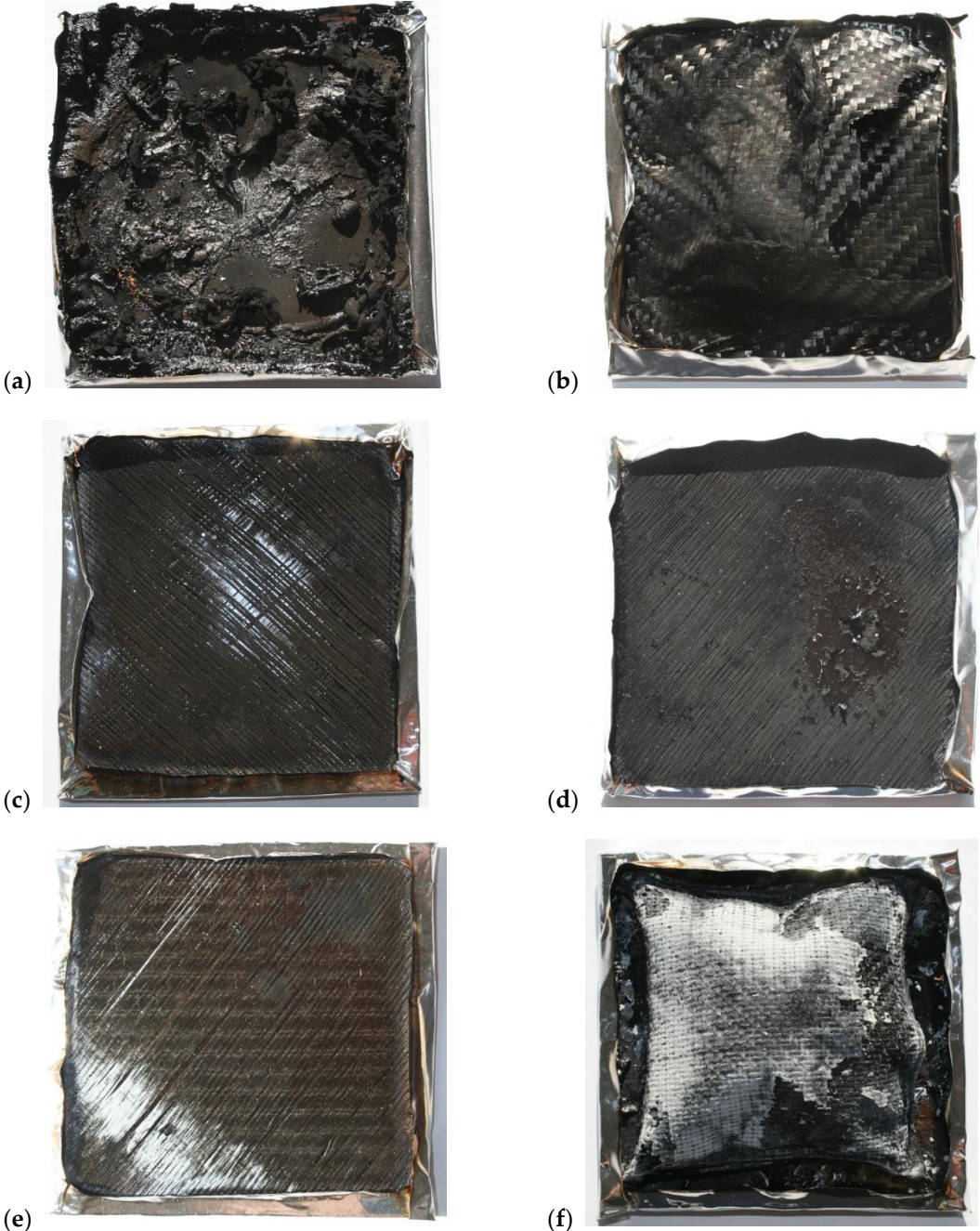

**Figure 14.** Photographs of EP (**a**), AF (**b**), GF (**c**), CF (**d**), BF (**e**), FF (**f**) after cone calorimeter (CC) tests.

## 4. Conclusions

Aramid, glass, carbon, basalt, and flax fibers were used to compare the epoxy-based composites' properties with similar grammage fabrics comprehensively. The proposed hand lay-up method, using six layers and the same amount of resin, allowed the formation of composites with the desired properties. Proper saturation of all fabrics by epoxy matrix and creating a coherent layer at the composite's surface positively impacted the performance. The highest mechanical properties characterized composites reinforced with aramid fabrics, which showed a tensile strength of 360 MPa and Young's modulus of 12.3 GPa, which adequately gave about 500% improvement of both parameters compared to epoxy resin. The lowest reinforcing efficiency was observed for glass fiber reinforced composites; however, GF composites also showed more than twice higher stiffness than EP.

All types of long fiber-reinforced composites showed increased impact strength compared to unmodified epoxy resin. The highest value was noted for BF reinforced composites, i.e., 127 kJ/m$^2$, which gives almost four times the EP value. The composites reinforced with natural fibers had the most unfavorable impact toughness.

The analysis of thermomechanical properties (DMTA) showed that the use of each type of fiber allows for a significant increase in the stiffness of composites at elevated temperature above the glass transition of the epoxy matrix. In the CF and BF reinforced composites, the storage modulus values at 140 °C amounted to approximately $5 \times 10^8$ Pa, which gives a value of almost one and a half orders of magnitude higher than unmodified EP.

Since the epoxy resin is the most flammable component, their lower amount in composites reduces burning and smoke emission intensity. The most promising results from composites were observed for resin with aramid fibers.

**Author Contributions:** Conceptualization, K.S.; methodology, K.S., M.B. and A.H.; formal analysis, K.S., M.B., J.A. and A.H.; investigation, K.S., M.B., J.A., A.H. and M.C.; resources, K.S.; writing—original draft preparation, K.S. and M.B.; visualization, K.S., M.B. and A.H.; project administration, K.S.; funding acquisition, K.S. All authors have read and agreed to the published version of the manuscript.

**Funding:** This paper has been based on the results of a research task carried out within the scope of the fifth stage of the National Programme "Improvement of safety and working conditions" partly supported in 2020–2022—within the scope of research and development—by the Ministry of Education and Science/National Centre for Research and Development. The Central Institute for Labour Protection–National Research Istitute is the Programme's main co-ordinator.

**Institutional Review Board Statement:** Not applicable.

**Informed Consent Statement:** Not applicable.

**Data Availability Statement:** Data available on request.

**Acknowledgments:** The study was realized with equipment allocated to Central Institute for Labour Protection–National Research Institute, as well as Poznan University of Technology and Gdansk University of Technology.

**Conflicts of Interest:** The authors declare no conflict of interest.

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
