# Peer review of "Comparative Study of the Reinforcement Type Effect on the Thermomechanical Properties and Burning of Epoxy-Based Composites"

_jcs, doi:10.3390/jcs5030089_

Round 1

Reviewer 1 Report

The present study on the comparison of different fabric reinforcements for epoxy polymers is interesting and it is supported with many characterization techniques. However, there are some points that need further attention. These points are described in the comments below:

  • The language of the text should be improved. Some expressions are not used in English, while there are some points that are difficult to understand (e.g., the preparation of composite samples).
  • Which were the dimensions of the produced laminate specimens? Were they consistent among the various samples (because it is mentioned that ~ 390 g of epoxy resin-curing agent mixture was used) or was a consistent additive/polymer weight ratio used? Many of the results are explained based on different volumetric content, so the weight and volume ratios of the composite systems must be clear to the reader.
  • Lines 109-110. The grammage of CF is 400 or 410 g/m2? Both values are mentioned.
  • In Lines 170-172 of FTIR discussion paragraph, it is stated that “additional peaks from reinforcing fillers are missing…”. It is suggested that FTIR spectra of neat reinforcing fillers should be also presented, proving the statement above.
  • In Figure 5, the chemical structures of CF and AF are inversely given. It should be corrected.
  • Comparison measurements of pristine epoxy polymer should be conducted for Tensile strength, Flexural strength, Impact strength and DMA tests.
  • In Figure 6, the Tensile strength results does not seem to correlate (as values, not as trend) with the presented stress-strain curves. Are the results the mean values of many measurements, while the presented curves were selected as the most representative to the results? If so, it should be stated. Furthermore, which strength values are presented? Maximum values or “at break” values?
  • In Lines 243 & 325, there are two references on the adhesion between the polymer matrix and the fillers. It must be stated if the authors are referring to increased interfacial interactions via chemical bonds (that should be visible with FTIR) or plain mechanical adhesion.
  • In Lines 324-326, the Tg values variation is trivial and no kind of interfacial interaction can be substantiated based on them.

Author Response

Dear Reviewer,

The Authors would like to thank you for your time and the review of the article. We corrected the manuscript according to your comments. All the changes in the text are marked with red font. We hope that the corrected paper will be suitable for publication in the Polymer Composites journal. Below please find our answers to your comments.

Yours sincerely K. Salasińska et al.

  • The language of the text should be improved. Some expressions are not used in English, while there are some points that are difficult to understand (e.g., the preparation of composite samples).

- The manuscript has been spell-checked and corrected in order to improve its legibility.

  • Which were the dimensions of the produced laminate specimens? Were they consistent among the various samples (because it is mentioned that ~ 390 g of epoxy resin-curing agent mixture was used) or was a consistent additive/polymer weight ratio used? Many of the results are explained based on different volumetric content, so the weight and volume ratios of the composite systems must be clear to the reader.

- Additional information about the amount of the filler in the composites has been included in 3.1. paragraph of the revised version of the manuscript.

  • Lines 109-110. The grammage of CF is 400 or 410 g/m2? Both values are mentioned.

- The "BIAX 400 g/m2" is a product trade name, which in fact, is characterized by the grammage of 411 g/m2.

  • In Lines 170-172 of FTIR discussion paragraph, it is stated that "additional peaks from reinforcing fillers are missing…". It is suggested that FTIR spectra of neat reinforcing fillers should be also presented, proving the statement above.

- The additional analysis of FTIR spectra of used reinforcing fibers has been presented in the revised manuscript version.

  • In Figure 5, the chemical structures of CF and AF are inversely given. It should be corrected.

- The correction was included in the revised manuscript version.

  • Comparison measurements of pristine epoxy polymer should be conducted for Tensile strength, Flexural strength, Impact strength and DMA tests.

- The additional samples made of pure epoxy resin were prepared and presented together with the composites' properties.

  • In Figure 6, the Tensile strength results does not seem to correlate (as values, not as trend) with the presented stress-strain curves. Are the results the mean values of many measurements, while the presented curves were selected as the most representative to the results? If so, it should be stated. Furthermore, which strength values are presented? Maximum values or "at break" values?

- In the case of stress-strain curves, the curves' courses are in good agreement with results presented in column plots. For both, tensile strength and elongation at break, are within the standard deviation of measured properties. Presented values are measured at the break for tensile test, while for a flexural test at strain calculated according to standard, the additional information was included in the revised manuscript version.

  • In Lines 243 & 325, there are two references on the adhesion between the polymer matrix and the fillers. It must be stated if the authors are referring to increased interfacial interactions via chemical bonds (that should be visible with FTIR) or plain mechanical adhesion.

- The additional information about plain mechanical adhesion, as the Reviewer suggested, was included in the manuscript's revised version.

  • In Lines 324-326, the Tg values variation is trivial and no kind of interfacial interaction can be substantiated based on them.

- The registered Tg conversion at the level of 6 °C is a significant change from the point of view and modification of the epoxy composition. However, the sentence was edited in order to exclude the possibility of raising any doubts.

Reviewer 2 Report

This manuscript represents the expeirmental analysis of the influence of the type of reinforcement on the thermomechanical performance of epoxy-based composites manufactured by hand-lay up methodology Five different reinforcements were analysed and different thermomechanical tests, such us DMTA, tensile, flexural and impact testing were carried out. The structure of the paper is well organized and the overall paper tells a logical story with a concrete conclusion. It is suitable for publication in Journal of Composites Science  after major revision.

1) Please explain in detail the novelty of this study and your scientific contribution.

2) Introduction and Reference sections should be improved with published works related to the analysis of thermomechanical and mechanical properties (tensile and flexural strength and stiffness, impact strength) of epoxy-based composites, as for example:

[1] I. García-Moreno, M.A. Caminero, G.P. Rodríguez, J.J. López-Cela, Effect of thermal ageing on the impact and flexural damage behaviour of carbon fibre-reinforced epoxy laminates, Polymers 11(1) (2019) 80

[2] I. García-Moreno, M.A. Caminero, G.P. Rodríguez, J.J. López-Cela, Effect of thermal ageing on the impact damage resistance and tolerance of carbon-fibre-reinforced epoxy laminates, Polymers 11(1) (2019) 160

3) Please include the basic mechanical properties of the epoxy resin and different reinforcements used in this study

4) Were the different manufactured plates analysed by ultrasonic or x-ray inspection in order to evaluate the manufacturing quality (inner defects, porosity, ...)

5) How many specimens were tested for each condition? Please include standard deviation of the experimental results.

Author Response

Dear Reviewer,

The Authors would like to thank you for your time and the review of the article. We corrected the manuscript according to your comments. All the changes in the text are marked with red font. We hope that the corrected paper will be suitable for publication in the Polymer Composites journal. Below please find our answers to your comments.

Yours sincerely K. Salasińska et al.

  • Please explain in detail the novelty of this study and your scientific contribution.

- The additional section, as well as comments related to motivation to prepare this work, was included in the introduction part. The work presented in this article constitutes preliminary research for the currently conducted works related to the production of hybrid composites modified using a new generation of flame retardants. The presented analysis aims to organize the research results and serve as a reference for further research. Also, as already mentioned, there is a significant amount of research on complex composite systems in the literature, and finding a different group of composites reinforced with one type of filler becomes a problem. This motivated us to prepare for this study.

2) Introduction and Reference sections should be improved with published works related to the analysis of thermomechanical and mechanical properties (tensile and flexural strength and stiffness, impact strength) of epoxy-based composites, as for example:

[1] I. García-Moreno, M.A. Caminero, G.P. Rodríguez, J.J. López-Cela, Effect of thermal ageing on the impact and flexural damage behaviour of carbon fibre-reinforced epoxy laminates, Polymers 11(1) (2019) 80

[2] I. García-Moreno, M.A. Caminero, G.P. Rodríguez, J.J. López-Cela, Effect of thermal ageing on the impact damage resistance and tolerance of carbon-fibre-reinforced epoxy laminates, Polymers 11(1) (2019) 160

- The additional references, including manufacturing of the epoxy composites using carbon fibers, with consideration of their mechanical and thermomechanical properties, have been included in a revised version of the manuscript.

3) Please include the basic mechanical properties of the epoxy resin and different reinforcements used in this study.

According to literature data, the mechanical properties of used fibers have been presented in Table 1, while the mechanical properties of unmodified epoxy resin were determined and included in the results section.

4) Were the different manufactured plates analysed by ultrasonic or x-ray inspection in order to evaluate the manufacturing quality (inner defects, porosity, ...)

- Unfortunately, due to the current pandemic situation, we do not have access to measuring equipment that would enable a non-destructive analysis of composites' structure. Nevertheless, the suggestion to use this type of analysis will certainly be used in further stages of the research work provided for in the project, which are the results discussed in this paper are the preliminary part.

5) How many specimens were tested for each condition? Please include standard deviation of the experimental results.

- The additional data have been presented in the revised version of the manuscript.

Reviewer 3 Report

General comments

The submitted manuscript deals with the characterization of composites with various reinforcement materials in terms of its thermomechanical and burning behavior. Although, the performed studies are typical in mechanical and thermomechanical characterization of composite materials and do not bring valuable insight in terms of methodology, the authors provided the comprehensive characterization of 5 composites that are commonly used in practice. This makes the submitted manuscript useful from the point of view of the performed comparisons and the technical information on the tested composites presented in a single study. Nevertheless, the manuscript needs substantial corrections, extensions, and explanations according to the detailed comments below.

Detailed comments

1) Line 40: it is recommended to substitute “thin-walled composites” with “thin-walled composite structures”, which is correct according to the widely accepted nomenclature.

2) Lines 46-47: the beginning of the sentence needs revision.

3) Line 51: I suppose the authors mean “E-glass” instead of “C-glass”.

4) Line 68: “are used primarily in the applications” or “are used in all applications”.

5) Lines 87-89: it is likely to present the mechanical properties in the form of a table to increase their readability and comparability.

6) In the Introduction, the authors focused mainly on the mechanical properties of various classes of composites, while the main research studies were performed in the area of thermomechanical properties and burning resistance. It is thus recommended to extend the Introduction by extending discussion in these topics.

7) It is also essential to mention about the demands to the investigated composite materials in terms of their thermomechanical properties and burning resistance and show their applicability and connection with industrial applications. I believe, this will help the authors to improve the formulation of the research problem.

8) The research problem is weakly formulated. I encourage the authors to reformulate it to show the originality of the research problem and the applicability of the obtained results (briefly).

9) In order to make the manufactured composites comparable, they should have similar manufacturing properties. The weights of fabrics of GF, CF, and BF are comparable, while the weights of AF and FF fabrics differ from the latter significantly. Please clarify, does the overall weight or volumetric contents of reinforcement is similar in the resulting composite structures? This also needs to be indicated in the manuscript.

10) Please double-check the manuscript for units other than SI units, and make appropriate changes.

11) Please comment on the applied parameters of the DMTA. Why torsion mode was considered? Is it due to the specific applications of the considered materials? Why the range od 25-200 deg.C was considered?

12) It would be useful to comment the results presented in Figure 3 also from the point of view of the reinforcing fiber types.

13) Figure 5: if the authors presented 4 of 5 chemical structures of the considered reinforcing fibers, the chemical structure of the basalt fiber can be also presented for consistency.

14) Lines 220-221: “Young's natural fiber composite modulus” should be replaced with “Young's modulus of the natural fiber composite”.

15) It would be useful to extend the discussion on fracture mechanisms presented on page 8 with the connections with fracture mechanics to improve understanding of the fracture mechanisms in the investigated composites.

16) Table 2: please correct the upper indices in units.

17) There is lack of discussion on the fire resistance results: the authors just stated the observations, which need to be enriched with the conclusions from these observations.

18) The conclusions of the manuscript need to be sufficiently extended, primarily by presenting the quantitative information from the performed studies.

Author Response

Dear Reviewer,

The Authors would like to thank you for your time and the review of the article. We corrected the manuscript according to your comments. All the changes in the text are marked with red font. We hope that the corrected paper will be suitable for publication in the Polymer Composites journal. Below please find our answers to your comments.

Yours sincerely K. Salasińska et al.

1) Line 40: it is recommended to substitute "thin-walled composites" with "thin-walled composite structures", which is correct according to the widely accepted nomenclature.

-The correction has been included in a revised version of the manuscript.

2) Lines 46-47: the beginning of the sentence needs revision.

- The sentence has been rewritten according to the Reviewers comment.

3) Line 51: I suppose the authors mean "E-glass" instead of "C-glass".

- The correction has been included in the revised version of the manuscript.

4) Line 68: "are used primarily in the applications" or "are used in all applications".

- The correction has been included in the revised version of the manuscript.

5) Lines 87-89: it is likely to present the mechanical properties in the form of a table to increase their readability and comparability.

- The mechanical properties of all used in this study fiber types have been presented in a table (Table 1).

6) In the Introduction, the authors focused mainly on the mechanical properties of various classes of composites, while the main research studies were performed in the area of thermomechanical properties and burning resistance. It is thus recommended to extend the Introduction by extending discussion in these topics.

7) It is also essential to mention about the demands to the investigated composite materials in terms of their thermomechanical properties and burning resistance and show their applicability and connection with industrial applications. I believe, this will help the authors to improve the formulation of the research problem.

8) The research problem is weakly formulated. I encourage the authors to reformulate it to show the originality of the research problem and the applicability of the obtained results (briefly).

- The additional section and comments related to motivation to prepare this work were included in the introduction part.

9) In order to make the manufactured composites comparable, they should have similar manufacturing properties. The weights of fabrics of GF, CF, and BF are comparable, while the weights of AF and FF fabrics differ from the latter significantly. Please clarify, does the overall weight or volumetric contents of reinforcement is similar in the resulting composite structures? This also needs to be indicated in the manuscript.

- Additional information about the amount of the filler in the composites has been included in 3.1. paragraph of the revised version of the manuscript.

10) Please double-check the manuscript for units other than SI units, and make appropriate changes.

- The suitable corrections of non-SI units and lack of adequate upper indexes have been included in the manuscript's revised version.

11) Please comment on the applied parameters of the DMTA. Why torsion mode was considered? Is it due to the specific applications of the considered materials? Why the range od 25-200 deg.C was considered?

- The use of DMA in the torsion mode results from the equipment available in our laboratory. A rotational rheometer equipped with an accessory for DMA measurements of solid samples was used for the tests. The choice of deformation geometry did not result from more or less favorable analytical effects. The used temperature range was selected to provide information on the possible post-curing effects of the polymer matrix. Moreover, the maximum temperature resulted from the thermal stability of the research materials confirmed in separate studies.

12) It would be useful to comment the results presented in Figure 3 also from the point of view of the reinforcing fiber types.

- The additional analysis of FTIR spectra of used reinforcing fibers has been presented in the revised manuscript version.

13) Figure 5: if the authors presented 4 of 5 chemical structures of the considered reinforcing fibers, the chemical structure of the basalt fiber can be also presented for consistency.

- The image of the chemical structure of basalt fiber has been included in the revised version of the manuscript.

14) Lines 220-221: "Young's natural fiber composite modulus" should be replaced with "Young's modulus of the natural fiber composite".

-The correction has been included in the revised version of the manuscript.

15) It would be useful to extend the discussion on fracture mechanisms presented on page 8 with the connections with fracture mechanics to improve understanding of the fracture mechanisms in the investigated composites.

- The additional comments related to the fracture mechanism of layered composites reinforced with long fibers were included in the part of the manuscript related to mechanical properties evaluation.

16) Table 2: please correct the upper indices in units.

- The values in Table 2 have been corrected.

17) There is lack of discussion on the fire resistance results: the authors just stated the observations, which need to be enriched with the conclusions from these observations.

 As indicated by the Reviewer, the analysis was performed only for comparative purposes, highlighting the problem of ascending differences in individual materials. It was nevertheless a deliberate act on the part of the authors. The work aimed to outline the different flammability of composites modified with different types of fibers. A broader discussion of the changes observed between the composites would require the introduction of additional measurements, including thermogravimetry or carbonaceous char analysis, which significantly exceeds the scope of the planned work. In subsequent research works, the assessment of the flammability of composites will be significantly expanded.

18) The conclusions of the manuscript need to be sufficiently extended, primarily by presenting the quantitative information from the performed studies.

- The conclusion has been modified by presenting briefly quantitative changes in composites, as recommended by the Reviewer.

Round 2

Reviewer 1 Report

No further comments are needed. 

Thank you for addressing all the issues pointed 

Reviewer 2 Report

The manuscript has been improved with the reply to the reviewer's comments. It is suitable for publication in Journal of Composites Science in present form.

Reviewer 3 Report

The authors addressed to all the provided comments and introduce corrections in the manuscript, significantly extending it from its previous version. All of the comments are satisfactory. The only technical issue is the correction of upper indices in the units. The manuscript can be considered for a publication in its present form.